# Nanomaterials for Periodontal Tissue Engineering: Chitosan-Based Scaffolds. A Systematic Review

**DOI:** 10.3390/nano10040605

**Published:** 2020-03-25

**Authors:** Dorina Lauritano, Luisa Limongelli, Giulia Moreo, Gianfranco Favia, Francesco Carinci

**Affiliations:** 1Department of Medicine and Surgery, Centre of Neuroscience of Milan, University of Milano-Bicocca, 20126 Milan, Italy; moreo.giulia@gmail.com; 2Interdisciplinary Department of Medicine, University of Bari, 70121 Bari, Italy; luisanna.limogelli@gmail.com (L.L.); gianfranco.favia@uniba.it (G.F.); 3Department of Morphology, Surgery and Experimental Medicine, University of Ferrara, 44121 Ferrara, Italy; crc@unife.it

**Keywords:** chitosan scaffold, periodontal tissue engineering, oral bone regeneration

## Abstract

***Introduction***. Several biomaterials are used in periodontal tissue engineering in order to obtain a three-dimensional scaffold, which could enhance the oral bone regeneration. These novel biomaterials, when placed in the affected area, activate a cascade of events, inducing regenerative cellular responses, and replacing the missing tissue. Natural and synthetic polymers can be used alone or in combination with other biomaterials, growth factors, and stem cells. Natural-based polymer chitosan is widely used in periodontal tissue engineering. It presents biodegradability, biocompatibility, and biological renewability properties. It is bacteriostatic and nontoxic and has hemostatic and mucoadhesive capacity. The aim of this systematic review is to obtain an updated overview of the utilization and effectiveness of chitosan-based scaffold (CS-bs) in the alveolar bone regeneration process. ***Materials and Methods***. During database searching (using PubMed, Cochrane Library, and CINAHL), 72 items were found. The title, abstract, and full text of each study were carefully analyzed and only 22 articles were selected. Thirteen articles were excluded based on their title, five after reading the abstract, twenty-six after reading the full text, and six were not considered because of their publication date (prior to 2010). Quality assessment and data extraction were performed in the twelve included randomized controlled trials. Data concerning cell proliferation and viability (CPV), mineralization level (M), and alkaline phosphatase activity (ALPA) were recorded from each article ***Results***. All the included trials tested CS-bs that were combined with other biomaterials (such as hydroxyapatite, alginate, polylactic-co-glycolic acid, polycaprolactone), growth factors (basic fibroblast growth factor, bone morphogenetic protein) and/or stem cells (periodontal ligament stem cells, human jaw bone marrow-derived mesenchymal stem cells). Values about the proliferation of cementoblasts (CB) and periodontal ligament cells (PDLCs), the activity of alkaline phosphatase, and the mineralization level determined by pure chitosan scaffolds resulted in lower than those caused by chitosan-based scaffolds combined with other molecules and biomaterials. ***Conclusions***. A higher periodontal regenerative potential was recorded in the case of CS-based scaffolds combined with other polymeric biomaterials and bioceramics (bio compared to those provided by CS alone. Furthermore, literature demonstrated that the addition of growth factors and stem cells to CS-based scaffolds might improve the biological properties of chitosan.

## 1. Introduction

Periodontal disease is a chronic inflammatory disease caused by a bacterial infection, which leads to an inflammatory status, causing destruction of the tissue supporting the teeth (the gingival, bone, and periodontal ligament) [1]. Periodontal inflammation resolution and subgingival microbial biofilm removal could only require nonsurgical mechanical therapy, but in order to obtain restitution ad integrum of the periodontum, alveolar bone defects need to be regenerated [2]. For this purpose, several biomaterials have been introduced, giving a positive impact on oral tissue engineering. These biomaterials act as three-dimensional scaffolds, the surface of which promotes cellular adhesion, proliferation, and differentiation, creating a favorable environment for tissue regeneration. Moreover, biomaterials, which can be of natural or synthetic origin, are able to come in immediate contact with the living tissue, without developing any adverse immune reaction. The placement of these novel biomaterials in the affected area activates a cascade of events, inducing regenerative cellular responses, and replacing the missing tissue [3,4]. The scaffold needs to have the capacity to promote osteogenesis, osteoinduction, and osteoconduction processes, cellular components (preosteogenic cells) must be delivered or attracted to its surface, and they must be activated by osteoinductive growth factors. Scaffold biomaterials are responsible for maintaining the appropriate space, in order to allow the implanted cells to deposit the ECM and to proliferate [5]. The recent review by Rodríguez-Vázquez et al. (2015) [6] specified the main characteristics of scaffold biomaterials that are used in tissue engineering. Biomaterials must be biocompatible, absorbable and degradable, with similar resorption and degradation rate (in vitro and in vivo) to the rate of tissue regeneration; its surface must be chemically adequate and stable; resistance and mechanical properties must be proper and its degradation products shall not be toxic or carcinogenic. The scaffold design should perfectly adapt to the defected area. An interplay between porosity and density should be present, as density improves mechanical strength and porosity facilitates cell migration, growth factors delivery, and vascularization [7,8]. Furthermore, the combined use of these biomaterials with mesenchymal stem cells (MSCs) and bone morphogenetic proteins (BMPs) with osteoinductive properties represents a crucial component of bone tissue engineering. MCSs present several features, which improve the regeneration process: they have immunosuppressive capacity, act as endocrine secretors, and are able to differentiate into several cellular types, such as osteoblasts and adipose cells [9]. The BMPs stimulate osteogenesis and neovascularization [5]. Jafari et al. (2015) [10] subdivided the scaffold biomaterials in two main groups: natural-based polymers and synthetic-based polymers, with different biodegradation rates, mechanical and physicochemical characteristics. The latter include polylactic acid (PLA), polyglycolic acid (PGA), and polylactic-*co*-glycolide (PLGA). Natural-based polymers are represented by chitosan, alginate, collagen, gelatin, elastin, and silk fibers. Natural polymers are perfectly biocompatibility and present several properties (pore size, porosity, fibrous structure), which guarantee positives results on living tissues. Chitosan (CS) is the fully or partially deacetylated form of chitin, which can be found in fungi and shells of sea crustaceans. It is the most abundant natural amino polysaccharide after cellulose and is formed by 2-acetamido-2-deoxy-β-d-glucopyranose and 2-amino-2-deoxy-β-d-glucopyranose groups, showing structural similarity to glycosaminoglycans of the extracellular matrix [11,12]. Chitosan is available in different forms, such as fibers, hydrogels, sponges, films and it is considered a valid biomaterial, capable of serving as two or three-dimensional scaffold in wound dressing or tissue engineering processes respectively. CS is a biodegradable, biocompatible, nontoxic, biologically renewable, and bacteriostatic biomaterial. The surface of chitosan is hydrophilic and it facilitates the adhesion, proliferation, and differentiation of the cellular component. Furthermore, it also has hemostatic and mucoadhesive capacity since its amino groups are positively-charged [13]. However, the studies by Alididi et al. and Marei et al. [14,15] demonstrated that chitosan was not osteoinductive or osteoconductive alone, but if combined with other molecules or biomaterials, such as growth factors, dental mesenchymal stem cells or hydroxyapatite, it could represent a valid help in bone regeneration.

### 1.1. Objectives

This research has the objective of reviewing the literature in order to obtain an updated overview of the utilization and effectiveness of chitosan-based scaffold (CS-bs) in the field of periodontal tissue engineering. This review focused, in particular, on CS-bs effectiveness in the alveolar bone regeneration process.

### 1.2. Clinical Question (PICO)

P: Chitosan-based scaffoldI: utilization and efficacy of chitosan-based scaffold in periodontal tissue engineering, assessing, in particular, its contribution in alveolar bone regenerationC: comparison between chitosan used alone and chitosan used in combination with other biomaterials, molecules or stem cellsO: general overview of the different chitosan scaffold forms and compositions and their application in periodontal tissue engineering. Evaluation of chitosan-based scaffold effectiveness in the alveolar bone regeneration process

## 2. Materials and Methods

### 2.1. Protocol and Registration

Methods and inclusion criteria of this systematic review were selected following the specific protocol provided by the PRISMA statement [16].

### 2.2. Eligibility Criteria

#### Inclusion and Exclusion Criteria

We have selected the most recent studies concerning periodontal tissue engineering, in which the alveolar bone regeneration was obtained by using a chitosan-based scaffold alone or in combination with other biomaterials, molecules or stem cells. The inclusion criteria were as follows:၀Available data about cell proliferation and viability, mineralization and alkaline phosphatase activity of the newly formed bone၀Study design: Randomized Controlled Trial၀Chitosan-based scaffold used in combination with other biomaterials, growth factors or stem cells၀Articles written in the English language

Case reports and reviews were excluded from our study. Studies published before 2010 were not considered.

### 2.3. Search

Databases of PubMed, Cochrane Library, and CINAHL were used to conduct a systematic literature review, selecting the most recent studies about the utilization and effectiveness of chitosan-based scaffold, applied to periodontal tissue engineering. Only articles written in the English language and published within 2010 were included. No restrictions were imposed regarding the type of biomaterials, molecules, or cellular components that were combined with chitosan. During literature searching, we used the following keywords: “chitosan scaffold” “periodontal engineering”, “oral bone regeneration” (combined with the Boolean term “AND”).

### 2.4. Study Selection and Data Collection Process

Study selection was conducted by two reviewers (D.L., G.M.), who independently reviewed title, abstract, and full text of all the articles that were found during the literature search. Eligible articles were selected following inclusion and exclusion criteria. Data collection was provided by two researchers (E.T., D.C.), who extracted from each item several pieces of information: the design of the study (randomized controlled trial), in vivo or in vitro analysis, and scaffold biomaterials that were used for bone regeneration. Data about cell proliferation and viability (CPV), mineralization (M), and alkaline phosphatase activity (ALPA) were recorded from each article. Therefore, the principal outcome measures referred to these parameters (means). The flow chart used for the selection of studies is shown in Figure 1.

### 2.5. Quality Assessment

Quality assessment of the included studies was provided by the Newcastle Ottawa Scale [17]. The lowest score was 5, the highest was 7, and on average the quality of the articles was evaluated to be equal to 6.04 (Table 1). All the items compared their results with a control group, and most of them were conducted in vitro. The potential for bone regeneration of each chitosan-based scaffold was analyzed with reliable methods, such as measurements of cell proliferation and viability (CPV), mineralization (M), and alkaline phosphatase activity (ALPA).

## 3. Results

### 3.1. Study Selection and Characteristics

Electronic research was conducted in PubMed, Cochrane Library, and CINAHL databases, and a total of 72 items were found. Title, abstract, and full text of each study were carefully analyzed and only 22 articles were selected. Thirteen articles were excluded based on title, five after reading the abstract, 26 after reading the full text, and six were not considered since they were published before 2010. After assessing the quality of the included articles using the Newcastle Ottawa Scale, they were submitted to data collection process. The twenty-two selected articles were randomized controlled trials, and all of them were written in the English language. Studies characteristics with reference to the author, in vivo/in vitro measurement, and type of biomaterial, are shown in Table 2, Table 3, Table 4, Table 5 and Table 6. In the selected studies, chitosan was used in combination with other biomaterials and molecules/cells. Biomaterials combined with CS were: hydroxyapatite (HA), alginate (AL), collagen, polylactic-co-glycolic acid (PLGA), pure polylactic acid (PLA), genipin, tricalcium phosphate, inorganic calcium phosphate, hyaluronic acid (Ha), dicarboxylic acid (DA), and polycaprolactone (PCL). Stem cells, growth factors, and proteins were also used in combination with CS: basic fibroblast growth factor (bFGF), periodontal ligament stem cells (PDLSCs), bone morphogenetic proteins (BMP), insulin-like growth factor-1 (IGF-1), osteoprotegerin (OPG), human jaw bone marrow-derived mesenchymal stem cells (hJBMMSCs) and human bone marrow stromal cells (hBMSCs). Ten of the included studies were performed in vitro, six of them were conducted in vivo and six both in vivo and in vitro. Studies in vivo used a sample of 48 mice (calvarial defects) and 12 beagles (alveolar bone). The included items analyzed several parameters, using different measurement methods. This study only reviewed data resulting from the evaluation of bone regeneration level, obtained thanks to the following measurement: (1) cellular proliferation and viability through MTT [40], Cell-Counting Kit-8®, AlamarBlue assays [41], and PrestoBlue assays (2) mineralization level using von Kossa, ARS (Alizarin Red S), Masson’s trichrome staining and immunofluorescent staining for osteocalcin (OCN), and (3) alkaline phosphatase activity (ALP).

### 3.2. Results of Individual Studies

The results of individual studies are presented in Table 2, Table 3, Table 4, Table 5 and Table 6. In order to evaluate the efficacy of chitosan–based scaffolds during bone regeneration, data about cell proliferation and viability were recorded from 10 of the included articles: five of them used the MTT assay of cementoblasts [18,19,20], periodontal ligament cells [18,28,37], one used the Cell-Counting Kit test (CCK-8) [23,26], four applied the AlamarBlue assay [25,29,30,32] and one the PrestoBlue test [22] (Table 2, Table 3 and Table 4). ALP activity of periodontal ligament cells [21], osteoblasts [34,35], and mesenchymal stem or stromal cells [24] was assessed in four studies (Table 5). Three items analyzed the mineralization level of the newly formed bone with the Masson’s trichrome staining method [27,33,38], two assessed it with the ARS staining [31,36] and one with the immunofluorescent staining technique for osteocalcin (OCN) [39] (Table 6). Twelve of the selected papers performed their experiment in vivo, creating bone defects, which were later covered by CS-based scaffolds: Ge et al. [21] created bilateral parietal bone defects (with a diameter of 5 mm) in eighteen eight-week-old rats (weight = 180–220 g), scoring the anesthetized cranial skin, exposing calvaria; parietal cranial 15 mm oval-shaped defects were obtained by Jayash et al. [25] using a bone trephine drill. A 5 mm diameter parietal defect was created in thirty-two five-month-old male rats by Li et al. [27] thanks to a trephine drill under copious saline irrigation. Guo et al. [23], after making an intraperitoneal injection with 10% chloral hydrate to 30 rats, performed a “V” type incision on the skull with a blade and drew with a drill, a 5 mm-diameter defect reaching the dura mater. Shah et al. [32] tested the CS-based scaffold on a subcutaneous pouch of eight-week healthy adult rats weighing 140–180 g. Xue et al. [36] anesthetize intramuscularly 3 white rabbits (4–6 months, weight 2–3 kg), exposing the lower edge of the mandible and creating a bone defect in the molar area of the mandibular body. Zang et al (2016) [38], used one-wall, box-shaped, infrabony defects (4 mm width, 7 mm depth) at the distal and mesial aspects of the third premolars and at the mesial aspects of the first molars. In the study of 2019 by the same author, bilateral class III furcation defects (4 mm wide and 5 mm high) were created on the third and fourth mandibular premolars [39]. The proliferation of cementoblasts (CB) and periodontal ligament cells (PDLCs) on pure chitosan scaffolds resulted in lower than those on the chitosan-based scaffolds combined with other molecules and biomaterials. The study by Akman et al. [18] compared the proliferation of these two cellular type on chitosan- based scaffold with the addition of HA and bFGF with those on pure chitosan one, showing that the absorbance values (at 570 nm) of the cells on day 7 and 8 were equal to 1.7 (CB) and 1 (PDLCs) and 0.6 (CB and PDLCs), respectively. Pure chitosan scaffolds were investigated in comparison with IGF-1, and BMP-6 added CS/AL/PLGA and β-tricalcium phosphate/CS scaffolds in the research by Duruel et al. [20] and Liao et al. [28] respectively, recording a higher absorbance value of CB and PDLCs in the second groups (2/2.6 and 0.9/1 on day 12 and 6, respectively). The MTT assay of PDLCs showed no significant differences between an autoclaved chitosan powder/β-glycerophosphate thermosensitive hydrogel (CS-PA/GP) and an autoclaved chitosan solution/GP hydrogel: absorbance values at 490 nm were equal to 0.7 and 0.6, respectively [37]. Dental pulp stem cells (DPSCs) were seeded on two scaffold types by Bakopoulou et al. [19]: CS combined with gelatine (CS/Gel) fabricated using 0.1% and 1% of the crosslinker glutaraldehyde (GTA). The OD values (545–630 mm) recorded in the MTT assay of DPSCs seeded on the two scaffolds after seven days were 1.6 for CS/Gel-0.1 and 1.3 for CS/Gel-1 (*p* < 0.01), but this statistical difference was compensated to non-significant at day 14. The trial by Miranda et al. [29] cultured osteoblast- and fibroblast-like cells on CS-Ha hydrogel, Ha hydrogel, and pure CS scaffolds; a quantitative evaluation of cell viability was conducted for 24, 48, and 72 h, using Alamar Blue, which was also added to a phosphate buffer solution without cells: the test showed increased cellular viability (20%) in both cellular groups compared with the control one; however, none of them were statistically different. The metabolic activity of hPDLCs and human bone marrow stromal cells (hBMSCs) recorded by the AlamarBlue assay in the trial by Mota et al. [30] presented higher values in the CS/bioactive glass nanoparticles (BG-NPs) membranes compared with CS one. The same test was performed by Jayash et al. [25] in order to assess the cell viability in a new osteoprotegerin-chitosan gel. After 24 h, the viability of the OPG-CS and CS gels (25 and 50 kDa) was significantly higher than those of the controls (OPG-CS and CS = 140%, controls = 110%). The cell proliferation and viability (assessed with AlamarBlue) of MC3T3-E1 cells on the trilayered functionally-graded CS membrane (FGM) with bioactive glass gradient (50%, 25%, 0% wt.) resulted in being higher than those in the control group: the relative percentage AB reduction after seven days was equal to 150% for tge FGM group and 90% for the control one [32]. In the in vitro study by Gümüşderelioğlu et al. [22], CS-based multifunctional and double-faced barrier membrane was realized: hard tissue was put in contact with the porous side of the membrane coated with HA, in which BMP-6 was also embedded. The nonporous surface of the membrane was in contact with the inflammatory soft tissue, and it was coated with electrospun PCL fibers. PrestoBlue assay on day 21 assessed that mitochondrial activities of MC3T3-E1 cells seeded on different membranes showed no statistical differences (CS = 0.58, HA/CS = 0.63, HA/CS + BMP-6 = 0.64, HA/CS/PCL = 0.62). Data demonstrated that these cells grew on all CS-based scaffolds, recording higher cellular activity in HA/CS membrane. The CCK-8 test of PDLCs in the article by Li et al. [26] demonstrated higher OD values (0.7) in CS-based hydrogel/α β –GP scaffold loaded with BMP2 plasmid DNA (pDNA-BMP2) than in those without pDNA-BMP2 (0.6). The same test used by Guo et al. [23] highlighted better cell viability on the electrospun collage-chitosan composite membrane than in the electrospun collagen one (OD values were 0.7 and 0.4 respectively). Sundaram et al. [34] analyzed the ability of a bilayered construct composed by PCL multiscale electrospun membrane and a chitosan/2 wt% CaSO_4_ scaffold to regenerate periodontal ligament and alveolar bone simultaneously. The authors of this study found a higher level of alkaline phosphatase activity of hDFCs on day 7 in the latter group (ALP protein concentration = 8 ng/mg) than in the control one (ALP protein concentration = 3.5 ng/mg). Multitissue simultaneous regeneration was also studied by Varoni et al. [35], who recorded no significant differences between the ALP activity of osteoblasts (OB) on day 7 provided by a CS-based genipin-cross-linked trilayered scaffold and the control group (460 and 480 pNpp/nmol min respectively). The RCT by Ge et al. [21] measured the ALP activity of PDLSCs up to 14 days in two different scaffold types: nanohydroxyapatite-coated –genipin-CS conjuction and genipin-CS-framework, showing higher values in the first group on day 7 (30 u/gprot and 25 u/gprot respectively), but registering similar values in both groups on day 14. The ALP activity of hMSCs seeded on HCG membrane recorded by Hunter et al. [24] showed a peak at 14 days of cultures, demonstrating that this type of membrane enhances hMSCs proliferation and osteogenic differentiation. Masson’s trichrome staining performed by Sukpaita et al. [33] found an increased amount of collagen and bone matrix in CS/Dicarboxylic acid scaffold with and without PDLCs seeding. In the study by Zang et al. [38], the same test showed more dense and well-organized PDLCs in the chitosan scaffold with hJBMMSCSs than the chitosan/anorganic bovine bone and pure chitosan groups. ARS staining of hPDLCs performed by Xue et al. [36] recorded more mineralized nodules on the nPLGA/nCS/nAG complex than in negative control group, showing the that this type of membrane may promote cell mineralization. Human mesenchymal stem cells (MSCs) were cultured by Rammal et al. [31] on a bone-mimetic material (B-MM) made from inorganic calcium phosphate combined with CS and hyaluronic acid biopolymers, which acted as a framework for the osteogenic potential of MSCs. ARS staining detected the formation by MMSCs of the mineralized matrix on B-MM, contrary to the control glass coverslip, on which no morphological changes and no nodules were found. In the study by Zang et al. [39], the number of OCN-positive cells in beagles mandibular class III furcation defects resulted in being higher on CS/β-GP/BMP-7/ORN and on CS/β-GP/BMP-7 membranes (45 and 43, respectively) than those on CS/β-GP/ORN and control group (22 and 19, respectively). Finally, Li et al. [27] used the Masson’s trichrome staining to compare the mineralization level of the newly formed bone (NB) in an injectable CS-based thermosensitive hydrogel scaffold with and without the incorporation of pDNA-BMP2 (CS/CSn(pDNA-BMP2)-GP). The study found out that the width of the NB was 500 µm for the first group and 300 µm for the second one, showing that CS/CSn-GP has greater capacity for alveolar bone regeneration when combined with pDNA-BMP2.

## 4. Discussion

This systematic review aimed to obtain an up-to-date overview of the usage and efficacy of chitosan-based scaffold used alone or combined with other biomaterials, (whose characteristics and properties are shown in Table 7 and Table 8), molecules, and cellular components. Thanks to its multiple properties, CS has been used for years in periodontal regeneration techniques [42,43,44,45,46]; the analysis of the most recent literature conducted in our paper highlighted that this biomaterial might be combined with other natural- or synthetic-based polymers obtaining bi- and trilayered scaffolds, which allow the simultaneous regeneration of the different tissues of the periodontal apparatus [21,25]. This capacity clarifies the reason why CS-based scaffolds should be used in the field of periodontal tissue engineering. CS is obtained from chitin deacetylation, which can be performed both through chemical or enzymatic processes: the chemical method avails of acids or alkalis, while the enzymatic one is made it possible by the chitin deacetylase, which catalyzes the hydrolysis of *N*-acetamido bonds in chitin [47]. Pure CS exists in various forms, depending on the molecular weights (300–1000 kDa) and on the degree of deacetylation, which generally ranges between 50–95%. These two parameters determine many physicochemical properties of CS, such as its solubility, crystallinity, and degradation. When the degree of deacetylation is intermediate, CS presents a semi-crystalline structure, while high deacetylation leads to a maximum crystallinity. The degradation rate of CS must provide the time necessary for the formation of the new bone: high degrees of deacetylation guarantee low degradation rates (which is performed in vivo by lysozyme). The free amine groups on deacetylated subunits present cationic nature, giving CS hemostatic, mucoadhesion, and antimicrobial properties. This biomaterial is biocompatible, biodegradable, and osteoconductive, facilitating the adhesion and proliferation of cells on its surface. The CS-based scaffold can be combined with other polymers and molecules in order to improve its mechanical and biological properties [48,49]. As a confirmation of this, in all the selected studies the data obtained from the analysis of the mineralization level, cell proliferation/viability, and alkaline phosphatase activity demonstrated that pure chitosan scaffolds were less effective at regenerating the bone tissue than the chitosan-based scaffolds combined with other biomaterials, molecules, and stem cells. Previous studies recorded superior mechanical reliability and in vivo biomineralization of CS combined with hydroxyapatite compared to CS and HA used alone [50]. According to the study by Akman et al. [18], the addition HA created a novel scaffold structure, preserving the pore sizes and interconnectivity. As well as decreasing the swelling ratio, it has been shown that HA established a strong mechanical interface with CS, which forms a hydrophilic structure, interacting with the body fluids. It was also showed that the higher was the chitosan concentration, the lower was the scaffold’s interconnectivity. In the same study, 100 ng of basic fibroblast growth factors were loaded to CS/HA scaffolds, demonstrating that the combination of these two biomaterials represents a superior carrier system for bFGF than CS alone. bFGF has the capacity to regulate periodontal wound healing, inducing the growth of immature PDL cells and also angiogenesis; it enhances the proliferation of osteoblasts, PDL cells, and cementoblasts. The addition of hydroxyapatite and the loading of basic fibroblast growth factor made possible for cementoblasts and periodontal ligament cells to increase their proliferation on the scaffold. The residual release of bFGF from the scaffold increased the proliferation of the cells, also thanks to its chemotactic effect [18,51]. Gümüşderelioğlu et al. [22] state that the presence of HA coating in CS membranes may lead to an increase of osteoconductivity of the scaffold. In the study by Ge et al. [21], the alkaline phosphatase activity of PDLSCSs resulted in being increased in nanohydroxyapatite coated scaffold: this may be caused by the release of calcium phosphate ions during the partial dissolution of nanohydroxyapatite. The small pores of this biomaterial also enhanced the attachment and proliferation of osteoblasts. Dental tissues mesenchymal stem cells as those obtained from the periodontal ligament may amplify the regenerative effect when seeded on scaffolds with proper surface characteristics: biodegradable polymer-nanohydroxyapatite composites may stimulate the differentiation of stem cells into osteoblasts. Rammal et al. [31] demonstrated that a bone-mimetic material made of organic CS combined with hyaluronic acid and calcium phosphate, may promote pro-regenerative secretome from MSCs since it represents a versatile osteoinductive coating. Dental pulp stem cells represent a precious option in regenerative dentistry [52], since they may potentiate the reconstitution of mineralized tissues, such as bone and dentine/pulp complex. For this reason, Bakopoulou et al. [19] seeded DPSCs on two CS-based scaffolds, which were combined with gelatin, fabricated with 0.1 and 1% of GTA. The combination with a gelatin may improve CS mechanical strength and its initial cell attachment potential. As well as several natural biomaterials, gelatin represents an attractive solution in tissue engineering, thanks to its biocompatibility, cell viability/proliferation maintenance, osteogenic differentiation promotions, and antimicrobial activity. The results of this study showed that the first day’s proliferation rate of DPSCs was lower in the scaffold with the highest concentration of GTA, a difference that disappeared at later time-points. Meanwhile, DPSCs seeded on CS/Gel-1 scaffold did not show upregulation of three differentiation markers, proving a long-term cytotoxic effect of GTA. It has been demonstrated that the combined action of bone morphogenetic proteins and scaffold polymers may enhance bone tissue regeneration. A study by Venkatesan et al. (2017) [53] proved that CS-based scaffolds have the capacity to systematically and sustainably release BMP-20 and Shu et al. [54] showed that the relationship between the osteogenic property of this protein and a 2-N,6-0-sulfated CS may enhance bone tissue development. Duruel et al. [20] highlighted the importance of growth factors in the regeneration process: IGF-1 promotes cell recruitment to the affected areas within a few hours, and BMPs are osteoinductive factors expressed in mature bone, which plays a crucial role in the regulation of bone metabolism. In this study, chitosan was combined with AL and PLGA microparticles. In order to analyze the effect of released growth factor on cellular functions, AL was used as a carrier for IGF-1 (which promotes OB proliferation and pre-osteoblasts differentiation), while PLGA as carrier for BMP-6 (that is an important biosignaling molecule in periodontal regeneration). AL was chosen because of its reversible swelling property, allowing growth factor release. PLGA is characterized by a low degradation rate, and it was demonstrated that it provided the growth factor release for a longer period than AL. Despite having many adequate properties, CS is nonbioactive but only biotolerable. This obstacle could be overcome, combining CS with calcium phosphates, which is bioactive and osteoconductive [28]. The association between CS and hyaluronic acid could be a valid option in periodontal tissue engineering: CS has better mechanical properties than Ha, but its bioresorption is longer than those of Ha [29]. Poor water solubility is one of the limitations of CS. Sukpaita et al. [33] prepared CS dissolving it in dicarboxylic acid, which also serves as crosslinking agents, giving higher mechanical properties to the chitosan scaffold. Periodontium is a complex structure, formed by different tissues (cementum, bone, periodontal ligament, and gingival). In order to obtain a multitissue simultaneous regeneration, the application of chitosan-based bylayered and trilayered scaffold seems to be a valid option [34,35]. The multilayered technique includes the use of a substrate, which is immersed in the CS solution, characterized by a cationic nature. In this way, the polymer deposits a thin film on the surface (layer). In order to realize the interaction between the positively charged groups of CS and the negative one, the system is immersed in a polyanionic solution. As a consequence, an upper layer is formed [55]. Bone marrow-derived mesenchymal stem cells may contribute to periodontal regeneration, as they can differentiate into cementum, bone, and periodontal ligament [38].

## 5. Conclusions

The efficacy of chitosan in periodontal tissue engineering has been widely demonstrated. Chitosan is a natural-based polymer with biodegradability, biocompatibility, and biological renewability properties. It is bacteriostatic and nontoxic and presents hemostatic and mucoadhesive capacity. In this systematic review, CS-based scaffolds combined with other polymeric biomaterials (AL, collagen, PLGA, PCL) and bioceramics (bioactive-glass, calcium phosphate, β-tricalcium phosphate, HA) have been analyzed, recording a higher periodontal regenerative potential compared to those given by CS alone. Data proved that natural biopolymers present high biocompatibility and antimicrobial potential; they have the capacity to recognize cell signals, to promote cell viability/proliferation and osteogenic differentiation. However, these biomaterials show weak mechanical characteristics. Synthetic polymers have poor potentiality in providing cell adhesion/migration and proliferation, but they show good mechanical properties, and their mechanical strength and degradation rate can be adjusted in order to reach the best performance. Bioceramics are bioactive, ensure excellent osteoconductivity and when combined with polymer matrix, they may improve the mechanical properties of the system. Furthermore, the addition of growth factors (bFGF, BMP, IGF-1) and stem cells (PDLSCs, hDFCs, hJBMMSCs) to CS-based scaffolds may improve the biological properties of chitosan, providing a better regenerative effect. Our analysis also highlighted that CS-based scaffolds accompanied by natural- or synthetic-based polymers might allow the simultaneous regeneration of the different periodontal tissues (gingival, cement, alveolar bone and periodontal ligament).

## Figures and Tables

**Figure 1 nanomaterials-10-00605-f001:**
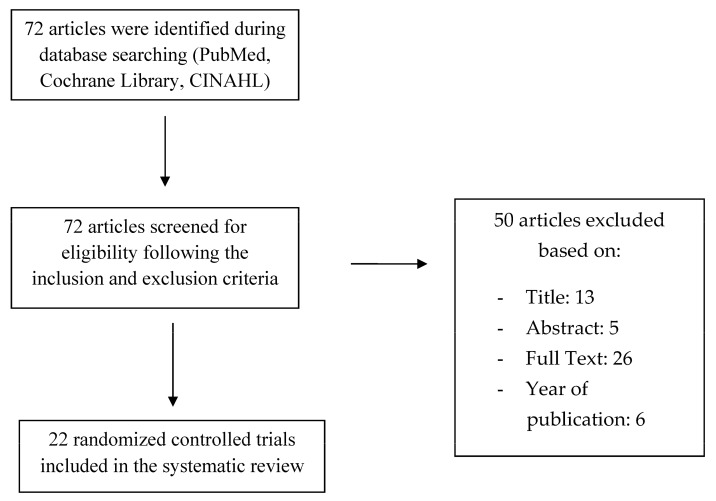
Flow chart of the publication assessment.

**Table 1 nanomaterials-10-00605-t001:** Quality assessment of included studies.

Studies	Definition of Cases	Representativeness of Cases	Selection of Controls	Definition of Controls	Comparability	Exposure	Total
[18]	+	+	-	-	+-	++-	6
[19]	+	+	+	+	+-	++-	7
[20]	+	+	-	-	+-	++-	6
[21]	+	+	+	-	+-	++-	6
[22]	+	+	+	+	+-	++-	7
[23]	+	+	+	+	+-	++-	7
[24]	+	+	-	-	+-	++-	5
[25]	+	+	-	-	+-	++-	5
[26]	+	+	-	-	+-	++-	6
[27]	+	+	-	-	+-	++-	5
[28]	+	+	+	-	+-	++-	6
[29]	+	+	-	-	+-	+++	6
[30]	+	+	+	+	+-	++-	7
[31]	+	+	+	+	+-	+--	6
[32]	+	+	-	-	+-	++-	5
[33]	+	+	+	-	+-	+++	7
[34]	+	+	-	-	+-	++-	5
[35]	+	+	+	-	+-	++-	6
[36]	+	+	+	+	+-	++-	7
[37]	+	+	+	-	+-	++-	6
[38]	+	+	+	-	+-	++-	6
[39]	+	+	+	-	+-	++-	6

+: star assigned; -: star not assigned.

**Table 2 nanomaterials-10-00605-t002:** Results of individual studies: MTT assay, absorbance values, 570 nm.

Studies	Type of Study	Biomaterials	Bone Regeneration Measurement	Results
[18]	In vitro	a) 2% and 3% (*w/v*) CS gel (Sigma-Aldrich, Germany) + 100 ng basic fibroblast growth factor (Sigma-Aldrich, Germany) b) 2% and 3% (*w/v*) CS (Sigma-Aldrich, Germany) combined with 1.25% (*w*/*v*) hydroxyhapatite (BioRad, USA) + 100 ng basic fibroblast growth factor (Sigma-Aldrich, Germany) c) CS (Sigma-Aldrich, Germany) combined with 1.25% (*w/v*) hydroxyapatite (BioRad, USA) d) CS alone (Sigma-Aldrich, Germany)	MTT assay of CB and PDLCs (absorbance values, 570 nm) at day 8 and 7 respectively	a) PDLCs = 0.9; CB = 0.85 b) PDLCs = 1; CB = 1.7 c) PDLCs = 0.6; CB = 0.5 d) PDLCs = O.6; CB = 0.6
[20]	In vitro	a) 1.1% (*w/v*) CS (Sigma-Aldrich Germany) combined with 2% (w/v) alginate (Fluka AG, Germany) and 2% (*w/v*) PLGA (Sigma-Aldrich, Germany) + 100 ng IGF-1 (R&D, USA) and 25 ng BMP-6 (USCN, China) b) 1.1% (*w/v*) CS (Sigma-Aldrich, Germany) combined with 2% (*w/v*) PLGA (Sigma-Aldrich, Germany) + 100 ng IGF-1 (R&D, USA) and 25 ng BMP-6 (USCN, China) c) CS (Sigma-Aldrich Germany) combined with alginate (Fluka AG, Germany) d) CS (Sigma-Aldrich Germany)	MTT assay of CB (absorbance values, 570 nm) at day 12	a) CB = 2.6 b) CB = 2.5 c) CB = 2.4 d) CB = 2
[28]	In vivo (left dorsal subcutaneous area in athymic mice)	a) CS (Sigma, St. Louis, MO, USA) dissolved in 2% acetic acid solution + β-tricalcium phosphate (Sigma–Aldrich Com) scaffold (mass ratio of β-tricalcium phosphate and CS = 3:7; total concentration = 1%) b) CS alone (Sigma, St. Louis, MO, USA)	MTT assay of PDLCs (absorbance values, 570 nm) at day 6	a) PDLCs = 1 b) PDLCs = 0.9
[37]	In vitro/in vivo (alveolar bone defects of mixed breed dogs)	a) CS solution (Sigma-Aldrich Chemical Co., St. Louis, MO, USA) and β-glycerolphosphate hydrogel (E. Merck, Darmstadt, Germany) b) CS powder and β-glycerolphosphate hydrogel (E. Merck, Darmstadt, Germany) c) Negative control	MTT assay of PDLCs (absorbance values, 490 nm) at day 6	a) PDLCs = 0.7 b) PDLCs = 0.6 c) PDLCs = 0.65

BMP-6 = bone morphogenetic protein-6; CB = cementoblasts; *IGF-1* = insulin-like growth factor-1; *MTT* = 3-(4, 5-dimethylthiazolyl-2)-2, 5-diphenyltetrazolium bromide; PDLCs = periodontal ligament cells; *PLGA* = polylactic-co-glycolic acid.

**Table 3 nanomaterials-10-00605-t003:** Results of individual studies: CCK-8 assay and MTT assay, OD values.

Studies	Type of Study	Biomaterials	Bone Regeneration Measurement	Results
[19]	In vitro	a) 2% (*w/v*) CS combined with 2% (*w/v*) gelatine, fabricated using 0.1% of glutaraldehyde b) 2% (*w*/*v*) CS combined with 2% (*w*/*v*) gelatine, fabricated using 1% of glutaraldehyde	MTT assay of DPSCs at day 7	a) OD value = 1.6 b) OD value = 1.3
[23]	In vitro/in vivo (calvarial bone defects of rats)	a) electrospun 5 wt% bovine tendon collagen membrane b) electrospun 5 wt% bovine tendon collagen-5 wt% chitosan membrane c) BLANK GROUP (no material applied at the cranial defect)	CCK-8 assay (Dojindo Ltd., Tokyo, Japan) of PDLCs	a) OD value = 0.4 b) OD value = 0.7 c) OD value = 0.22
[26]	In vitro	a) CS (Qingdao Boyite Biomaterials Co. Ltd., Shandong, China) combined with CSn (pDNA-BMP2)-GP^(^*) b) CS (Qingdao Boyite Biomaterials Co. Ltd., Shandong, China) combined with CSn-GP^(*)^	CCK-8 assay (Dojindo Ltd., Tokyo, Japan) of PDLCs	a) OD value = 0.7 b) OD value = 0.6

*CCK-8* = cell counting kit-8; *CSn(pDNA-BMP2)-GP* = chitosan nanoparticles loaded with bome morphogenetic protein-2 plasmid DNA into a chitosan-based hydrogel with α, β–glycerophosphate; *DPSCs = dental pulp stem cells; tudies: CCK-8 assay and MTT assay, OD values. DPSCs* = dental pulp stem cells; *MTT* = 3-(4,5-dimethylthiazolyl-2)-2,5-diphenyltetrazolium bromide; *OD* = optical density; *PDLCs* = periodontal ligament stem cells. (*) *CS for nanoparticles*: Hengtai Jinhu Crust Product Co. Ltd. (Shandong, China); *αβ-GP*: Sinopharm Chemical Reagent Co. Ltd.; *Plasmid BMP2*: Central Laboratory of the Affiliated Hospital of Qingdao University.

**Table 4 nanomaterials-10-00605-t004:** Results of individual studies: AlamarBlue and PrestoBlue assay.

Studies	Type of Study	Biomaterials	Bone Regeneration Measurement	Results
[22]	In vitro	a) CS alone (Sigma-Aldrich, Germany) b) 1.1% (*w/v*) CS (Sigma-Aldrich, Germany) combined with hydroxyapatite c) CS (Sigma-Aldrich, Germany) combined with hydroxyapatite and 1% (*w/v*) PCL (Sigma-Aldrich, Germany) solution in HFIP (Sigma-Aldrich, Germany) d) CS (Sigma-Aldrich, Germany) combined with hydroxyapatite + 30 μL of BMP-6 solution including 100 ng of BMP-6 (recombinant human, R&D Systems, Minneapolis, MN, USA)	PrestoBlue assay of MC3T3-E1 cells (absorbance values, 570 nm) at day 21	a) MC3T3-E1 = 0.72 b) MC3T3-E1 = 0.65 c) MC3T3-E1 = 0.67 d) MC3T3-E1 = 0.66
[25]	In vitro/in vivo (calvarial bone defects of rabbits)	a) 50 kDa CS b) 25 kDa CS c) 10 kDa CS d) 50 kDa CS + 1 mg/mL^−1^ osteoprotegerin (PeproTech, Rocky Hill, NJ, USA) e) 25 kDa CS + 1 mg/mL^−1^ osteoprotegerin (PeproTech, Rocky Hill, NJ, USA) f) 10 kDa CS + 1 mg/mL^−1^ osteoprotegerin (PeproTech, Rocky Hill, NJ, USA) g) Control	AlamarBlue assay (Sigma) of osteoblasts (metabolic viability %) seeded on the gels after 24 h	a) Viability = 140% b) Viability = 120% c) Viability = 120% d) Viability = 150% e) Viability = 140% f) Viability = 150% g) Viability = 110%
[29]	In vitro	a) 0.5g CS (Sigma Aldrich, St. Louis, MO, USA) + 1 g hyaluronic acid hydrogel (Sigma Aldrich, St. Louis, MO, USA) scaffold b) 1% (*w*/*v*) CS (Sigma Aldrich, St. Louis, MO, USA) hydrogel scaffold c) Ha hydrogel (Sigma Aldrich, St. Louis, MO, USA) scaffold seeded with osteoblast and fibroblast cultures PBS solution without cells (negative control group)	AlamarBlue assay (mitochondrial cell activity)	The test showed a higher increase (20%) in cellular viability of both cellular groups compared with the control one. However, none of them were statistically different
[30]	In vitro	a) 0.7% (*w/v*) purified CS (Sigma-Aldrich) combined with 0.3% (*w/v*) bioactive-glass nanoparticles in a solution of 2 vol% acid acetic b) CS alone (Sigma-Aldrich)	AlamarBlue assay of PDLCs and hBMSCs	Generally cell metabolic activity was higher in the CS/BG-NP group compared with the CS group for both PDLCs and hBMSCs.
[32]	In vitro/in vivo (8 weeks old rats)	a) Trilayered functionally-graded chitosan membrane (FGM) with bioactive glass gradient (50%, 25%, 0% wt.) b) control group	AlamarBlue assay of MC3T3-E1 cells (relative % AB reduction)	a) Relative % AB reduction = 150% b) Relative % AB reduction = 90%

*BMP-6* = bone morphogenetic protein-6; *CS/BG-NP* = chitosan/bioactive-glass nanoparticles; *FGM* = functionally-graded chitosan membrane; Ha = hyaluronic acid; *hBMSCs* = human bone marrow stromal cells; *HFIP* = 1,1,1,3,3,3-hexafluoro-2 propanol*; PBS* = phosphate buffer solution; *PCL* = polycaprolactone; PDLSCs = periodontal ligament stem cells;

**Table 5 nanomaterials-10-00605-t005:** Results of individual studies: ALP activity.

Studies	Type of Study	Biomaterials	Bone Regeneration Measurement	Results
[21]	In vivo (calvarial bone defects of rats)	a) nanohydroxyapatite-coated genipin-chitosan conjunction scaffold + 1 × 10^7^/mL PDLSCs b) genipin-chitosan framework + 1 × 10^7^/mL PDLSCs	ALP activity of PDLSCs (U/gprot = unit/gram protein) at day 7	a) ALP activity = 30 U/gprot b) ALP activity = 25 U/gprot
[24]	In vitro	a) 0.4g hydroxyapatite-0.5g chitosan-0.5g gelatin (Sigma-Chemical Co. St. Louis, MO) + hMSCs	ALP activity of hMSCs (µmol/(10^6^ cells x min) at day 14	a) µmol/(106 cells x min = 2.4
[34]	In vitro	a) Bilayered construct consisting of PCL (Poly Sciences, Warrington, PA) multiscale electrospun membrane + 2 g CS (Koyo Chemical, Japan) hydrogel/2 wt% CaSO_4_ (Fischer Scientific, USA) scaffold b) CS control(Koyo Chemical, Japan) scaffold	ALP to evaluate the differentiation of hDFCs to OB at day 7	a) ALP protein concentration = 8 ng/mg b) ALP protein concentration = 3.5 ng/mg
[35]	In vitro/in vivo	a) CS- based trilayer porous scaffold b) Control scaffolds	ALP assay of OB on the different compartments after day 7	a) OB = 460 pNpp/nmol min b) OB = 480 pNpp/nmol min

*ALP* = alkaline phosphatase activity; *CS* = chitosan*; hDFCs* = human dental follicle stem cells; *hMSCs* = human mesenchymal stem cells; *OB* = osetoblasts; *PCL* = polycaprolactone; *PDLSCs* = periodontal ligament stem cells.

**Table 6 nanomaterials-10-00605-t006:** Results of individual studies: Masson’s trichrome staining, ARS staining, OCN staining.

Studies	Type of Study	Biomaterials	Bone Regeneration Measurement	Results
[27]	In vitro/in vivo (male rats)	a) CS (Qingdao Boyite Biomaterials Co. Ltd., Shandong, China) combined with CSn(pDNA-BMP2)-GP^(*)^ b) CS (Qingdao Boyite Biomaterials Co. Ltd., Shandong, China) combined with CSn-GP(*)	Masson’s trichrome staining (width of new bone)	a) NB = 500 µm b) NB = 300 µm
[33]	In vivo (calvarial bone of mice)	a) CS combined with dicarboxylic acid + PDLCs b) CS combined with dicarboxylic acid without PDLCs	Masson’s trichrome staining	Increase in the amount of collagen and bone matrix in CS/DA scaffold with and without PDLCs after 12 weeks
[38]	In vivo (alveolar bone of male beagles)	a) CS anorganic bovine bone b) CS anorganic bovine bone + hJBMMSCs c) CS d) CS + hJBMMSCs e) anorganic bovine bone f) Control group	Masson’s trichrome staining	CS + hJBMMSCs showed more dense and well-organized PDL than the other groups
[31]	In vitro	a) CS combined with inorganic calcium phosphate and hyaluronic acid on which MSCs were cultured b) control glass coverslip	ARS staining of MSCs	a) Histological and immunohistochemical analysis of paraffin-embedded nodules evidenced the presence of mineralized matrix positive to red alizarin and cells positive to osteocalcin. b) No major morphological changes and no nodules were observed
[36]	In vivo (molar area of the mandibular body of rabbits)	a) PLGA nanoparticles (Shandong Institute of Medical Instruments) (100 mg PLGA dissolved in 10 mL acetone and slowly poured into 40 mL 2% polyvinyl alcohol) /CS nanoparticles (Tiengene Bio-Technique Co. Ltd., Guangzhou, China) (40 mg CS powder dissolved in 40 mL of 1% glacial acetic acid, then 10 mL of 0.1% TPP solution instilled into the CS solution)/Silver nanoparticles complex (Shanghai Chaowei Nanotechnology Co. Ltd. DMEM (HyClone)	ARS staining of PDLCs	More mineralized nodules were observed on nPLGA/nCS/nAG membrane than in negative control group
[39]	In vivo (alveolar bone of male beagles)	a) 200 mg CS (Sigma-Aldrich, St. Louis, MO, USA) combined with 56% *w*/*v*, 1 mL β-glycerophosphate (Merck, Darmstadt, Germany) thermosensitive hydrogel + 100 ng/mL BMP-7 (PeproTech, Rocky Hill, NJ, USA) and 0.5% ornidazole b) 200 mg CS (Sigma-Aldrich, St. Louis, MO, USA) combined with56% *w*/*v*, 1 mL β-glycerophosphate (Merck, Darmstadt, Germany) thermosensitive hydrogel + 100 ng/mL ornidazole c) CS (Sigma-Aldrich, St. Louis, MO, USA) combined with 56% *w*/*v*, 1 mL β-glycerophosphate (Merck, Darmstadt, Germany thermosensitive hydrogel + 100 ng/mL BMP-7 (PeproTech, Rocky Hill, NJ, USA) d) CS alone (Sigma-Aldrich, St. Louis, MO, USA)	OCN staining for osteoblasts (number of OCN-positive cells)	a) OCN-positive cells = 45 b) OCN-positive cells = 22 c) OCN-positive cells = 43 d) OCN-positive cells = 19

*AG =* silver; ARS = *ARS* = alizarine red; *BMP-7* = bone morphogenetic protein-7; CSn*(pDNA-BMP2)-GP* = chitosan nanoparticles loaded with bone morphogenetic protein-2 plasmid DNA into a chitosan-based hydrogel with α, β–glycerophosphate; DA = dicarboxylic acid; *NB* = new bone; *hJBMMSCs* = human jaw bone marrow-derived mesenchymal stem cells*; MSCs =* mesenchymal stem cells; *PDLCs* = periodontal ligament cells. ^(*)^
*CS for nanoparticles*: Hengtai Jinhu Crust Product Co. Ltd. (Shandong, China); *αβ-GP*: Sinopharm Chemical Reagent Co. Ltd.; *Plasmid BMP2*: Central Laboratory of the Affiliated Hospital of Qingdao University

**Table 7 nanomaterials-10-00605-t007:** Bioceramics properties.

BIOCERAMICS	CHACARCTERISTICS and PROPERTIES
HYDROXYAPATITE	- Biocompatible - Bioactive - Osteoinductive and osteoconductive properties - It provides excellent mechanical strength to the scaffold - Nanoparticles of HA presents better delivery mechanism
CALCIUM PHOSPHATE	- Biocompatible - Bioactive - High mechanical properties - Good osteoconductive properties - High bio-resorption - Support of cells (osteoblasts, mesenchymal cells) attachment, differentiation and proliferation
BIOACTIVE GLASS	- Biocompatible - Bioactive - Biomineralization capability - Strong chemical bond with the host bone tissue - Angiogenesis induction capacity - Osteoconductive and osteoproductive properties

**Table 8 nanomaterials-10-00605-t008:** Natural and synthetic biopolymers properties.

BIOPOLYMER	CHARACTERISTICS and PROPERTIES
ALGINATE	- Biocompatible - Biodegradable - Hydrophilic under physiological conditions - Capacity to encapsulate living cells - Poor mechanical properties - Uncontrolled degradation under physiological condition
COLLAGEN	- Biocompatible - Biodegradable - High tensile strength - High affinity with water - Good osteogenesis properties - Poor mechanical properties - Osteoblasts adhesion promotion
HYALURONIC ACID	- Biocompatible - Poor mechanical strength - Rapid bio-resorption - Hydrophilic - Poor mechanical properties - Cells proliferation, movement and differentiation regulation capacity - Stem cells differentiation capacity - Poor cell adhesion - Molecular carrier
POLYCAPROLACTONE (PCL)	- Biocompatible - Bioabsorbable - Slow degradation rate - Hydrophobic - Viscoelastic properties - Good mechanical properties - Poor cell adhesion and proliferation capacity
POLYLACTIC-CO-GLYCOLIDE (PLGA)	- Biocompatible - Slow degradation rate - Nontoxic degradation metabolites - Relatively hydrophobic - Good mechanical properties - Thermal processibility - Molecular carrier

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
