# Peer review of "Nanomaterials for Periodontal Tissue Engineering: Chitosan-Based Scaffolds. A Systematic Review"

_nanomaterials, 2020, doi:10.3390/nano10040605_

Round 1
Reviewer 1 Report
Using Chitosan in different dental disciplines because of its non toxicity and bioactivity is promising. Indications comprise healing treatment of different tissues such as bone, pulp, periodontal ligament. This systematic review gives very good insight in possibilities and problems in bone regeneration in parodontology. Literature used is up to date and analytic process is well designed. Conclusions are based on literature analysis.
Author Response
Milan, 2nd March 2020
Dear editor:
Many thanks for the insightful comments and suggestion of the referees. We have
made corresponding revision according to their advice. Words in red are the changes
we have made in the text.
Thank you for receiving our manuscript and considering it for publication.
We appreciate your time and look forward to your response.
Yours sincerely,
Dorina Lauritano
Reviewer 2 Report
This paper reports systematic review of chitosan-based scaffolds for periodontal tissue engineering. The novelty in this study is the effect of chitosan-based materials for periodontal tissue engineering. This work is, in general, organized and presented very well. However, the central result and finding are rather weak, because the novelty of chitosan (or -based scaffolds) is not so strong. Authors should be clarify the reason why chitosan should apply to periodontal tissue engineering field. Also, the Table 2 and 3 should be re-arranged to be easy to see for readers. Furthermore, author should add more enough references to prove the logic of the study.
Author Response
Milan, 2nd March 2020
Dear editor:
Many thanks for the insightful comments and suggestions of the referees. We have
made corresponding revision according to their advice. Words in red are the changes
we have made in the text.
The revisions are as follows:
- The central result and finding are rather weak, because the novelty of chitosan (or -based scaffolds) is not so strong. Authors should be clarify the reason why chitosan should apply to periodontal tissue engineering field.
We have extended the “Discussion” section, explaining the reason why chitosan should be applied to periodontal tissue engineering field: “Thanks to its multiple properties, CS has been used for years in periodontal regeneration techniques [40,41,42,43,44] ; the analysis of the most recent literature conducted in our paper highlighted that this biomaterial may be combined with other natural- or synthetic-based polymers obtaining multilayered scaffolds, which allow the simultaneous regeneration of the different tissues of the periodontal apparatus [27,31]. This capacity clarifies the reason why CS-based scaffolds should be used in the field of periodontal tissue engineering”.
- Table 2 and 3 should be re-arranged to be easy to see for readers.
We have modified Table 2 and 3, facilitating the reading.
- Author should add more enough references to prove the logic of the study.
We included 2 more articles in the review and we have added new references
Thank you for receiving our manuscript and considering it for publication.
We appreciate your time and look forward to your response.
Yours sincerely,
Dorina Lauritano

Reviewer 3 Report
The manuscript entitled "Nanomaterials for periodontal tissue engineering: chitosan-based scaffolds. A systematic review." is write more like a technical report, rather than a review.
One major issue is the fact that there is no figure reproduced from other published works. This is a mandatory subject in order to provide valuable information for readers.
Table 1 has no mining since is was only the judgement of the authors. Usually, a review is comprehensive literature study in order to provide valuable information for readers related to a certain subject. In this case, is more an evaluation of the published studies.
A similar discussion could be made for Figure 1.
In order to reconsider the review, the authors must present data from previously published researches related to the domain, including graphical information and also correlations and comment related to presented papers in the manuscript.
Regarding the references, 42 citations for ‘a systematic review’ are not enough.
A systematic review with less than 90-100 citations, is not systematic, is ‘mini review’.
I am convinced that the authors could revise carefully their manuscript and provide a better manuscript after major revision.
Author Response
Milan, 2nd March 2020
Dear editor:
Many thanks for the insightful comments and suggestions of the referees. We have
made corresponding revision according to their advice. Words in red are the changes
we have made in the text. We have re-organized our paper following the PRISMA statement for systematic review.
Thank you for receiving our manuscript and considering it for publication.
We appreciate your time and look forward to your response.

Round 2
Reviewer 2 Report
Dear. Authors
Thank you for the great re-submission with collect revised version.
Author Response
Milan, 11th March 2020
Dear Reviewer,
Thank you for appreciating our re-submission.
Yours sincerely,
Dorina Lauritano

Reviewer 3 Report
One major issue is the fact that there is no figure reproduced from other published works. This is a mandatory subject in order to provide valuable information for readers.
One missing aspect of the review is the discussion about the animals used in experiments. According the table 2, different animals was used (rats, mice, dogs, rabbits, beagles) and a discussion about this aspect is mandatory.
I think that the words “multilayered scaffolds” (line 310) must be reformulated because is not correct.
Regarding the discussion about CS-based scaffolds combined with other biomaterials, I expect to find some information about the ratio of these biomaterials. What is the optimal percent of hyaluronic acid or hydroxyapatite when we want to add these biomaterials on CS-based scaffolds, according to the studies analyzed by the authors?
The authors must reformulate the conclusion section. I think that the authors must provide in conclusion section a clear vision related to the combination between CS and ceramic biomaterials versus CS and polymeric biomaterials. Also, is better to clarify the influence of the growth factors and stem cells on CS-based scaffolds. According the existing sentence, the idea that growth factors or stem cells will improve the mechanical properties of CS-based scaffolds is not real. (“However, the combination between CS and other biomaterials (HA, AL, PLGA, PLA, PCL), growth factors (bFGF, BMP, IGF-1) and stem cells (PDLSCs, hDFCs, hJBMMSCs) may improve its mechanical and biological properties, providing a better regenerative effect.”)
I am convinced that the authors could revise carefully their manuscript and provide a better manuscript after major revision.
Author Response
Milan, 11th March 2020
Dear editor,
Many thanks for the insightful comments and suggestion of the referees. We have
made corresponding revision according to their advice. Words in red are the changes
we have made in the text.
- One major issue is the fact that there is no figure reproduced from other published works. This is a mandatory subject in order to provide valuable information for readers.
We reported some figure from other published works.
- One missing aspect of the review is the discussion about the animals used in experiments. According the table 2, different animals was used (rats, mice, dogs, rabbits, beagles) and a discussion about this aspect is mandatory.
We added the description of in vivo experiments in the “Results” section.
- I think that the words “multilayered scaffolds” (line 310) must be reformulated because is not correct.
We have corrected the words “multilayered scaffolds”, replacing it with the words “bi- and trilayered scaffolds”
- Regarding the discussion about CS-based scaffolds combined with other biomaterials, I expect to find some information about the ratio of these biomaterials. What is the optimal percent of hyaluronic acid or hydroxyapatite when we want to add these biomaterials on CS-based scaffolds, according to the studies analyzed by the authors?
We have added information about biomaterials ratio and concentrations in Table 2.
- The authors must reformulate the conclusion section. I think that the authors must provide in conclusion section a clear vision related to the combination between CS and ceramic biomaterials versus CS and polymeric biomaterials. Also, is better to clarify the influence of the growth factors and stem cells on CS-based scaffolds. According the existing sentence, the idea that growth factors or stem cells will improve the mechanical properties of CS-based scaffolds is not real. (“However, the combination between CS and other biomaterials (HA, AL, PLGA, PLA, PCL), growth factors (bFGF, BMP, IGF-1) and stem cells (PDLSCs, hDFCs, hJBMMSCs) may improve its mechanical and biological properties, providing a better regenerative effect.”).
We corrected the “Conclusion” section.
Thank you for receiving our manuscript and considering it for publication.
We appreciate your time and look forward to your response.
Yours sincerely,
Dorina Lauritano

Round 3
Reviewer 3 Report
The authors perform all required modifications and the manuscript is suitable for publication.
Author Response
Milan, 15th March 2020
Dear Reviewer,
Thank you for appreciating our re-submission.
Yours sincerely,
Dorina Lauritano
